

# Inducible nitric oxide synthase (NOS2) knockout mice as a model of trichotillomania

Plinio C. Casarotto[1,2], Caroline Biojone[1,3], Karina Montezuma[3], Fernando Q. Cunha[2], Samia R.L. Joca[3], Eero Castren[1] and Francisco S. Guimaraes[2,4]

[1] Neuroscience Center, University of Helsinki, Helsinki, Finland
[2] Department of Pharmacology, Ribeirão Preto Medical School, University of São Paulo, Ribeirão Preto, São Paulo, Brazil
[3] Department of Physics and Chemistry, School of Pharmaceutical Sciences of Ribeirão Preto, University of São Paulo, Ribeirão Preto, São Paulo, Brazil
[4] NAPNA: Center for Interdisciplinary Research on Applied Neurosciences, University of São Paulo, Ribeirão Preto, São Paulo, Brazil

## ABSTRACT

**Background**. Trichotillomania (TTM) is an impulse control disorder characterized by repetitive hair pulling/trimming. Barbering behavior (BB) observed in laboratory animals is proposed as a model of TTM. The neurobiological basis of TTM is unclear, but involves striatal hyperactivity and hypoactivation of the prefrontal cortex.

**Methods**. In this study, we evaluated the BB in knockout mice for the inducible isoform of nitric oxide synthase (NOS2KO) and the consequences of silencing this enzyme in PC12 cell differentiation.

**Results**. NOS2KO exhibit exacerbated BB, starting four weeks of age, and increased repetitive movements compared to wild-type mice (WT). The expression of BB was attenuated by repeated treatment with clomipramine, a clinically approved drug to treat TTM in humans, or memantine, an antagonist of NMDA receptors, as well as partial rescue of NOS2 expression in haploinsufficient animals. The silencing of NOS2 expression reduced the MAP2 (microtubule-associated protein 2) levels in activity-induced differentiated PC12 cells.

**Discussion**. Our data led us to propose that NOS2 is putatively involved in the neuronal maturation of the inhibitory afferent pathways during neurodevelopment, and such inadequate inhibition of motor programs might be associated to the observed phenotype.

Corresponding author
Plinio C. Casarotto,
plinio.casarotto@helsinki.fi

## INTRODUCTION

The cortico-striato-thalamo-cortical circuitry (CSTC) is a series of reverberatory loops that control motor programs. The glutamatergic neurotransmission plays a crucial role in both afferent and efferent pathways in prefrontal cortex and striatum, two core structures in the regulation of CSTC (*Langen et al., 2011a*; *Langen et al., 2011b*). In this scenario, nitric oxide (NO) plays a major part, not only regulating the release of glutamate but also in

the migration and maturation of neuronal cells in the cortex. This later feature of NO is apparently mediated by the inducible isoform of nitric oxide synthase (NOS2) (*Arnhold et al., 2002*).

Malfunctions of specific parts of the CSTC are speculated to be involved in the neurobiology of several motor and impulse control disorders, such as obsessive-compulsive disorder, Tourette's syndrome, and of interest to the present study, trichotillomania (*Langen et al., 2011b*; *Langen et al., 2011a*).

Trichotillomania (TTM) is an impulse control disorder characterized by a distressing urge for hair pulling or trimming, affecting 1–3.5% of the population (*Christenson, Pyle & Mitchell, 1991*; *American Psychiatric Association, 2013*). Nowadays, the pharmacotherapy for TTM is based on treatment with the serotonin reuptake inhibitor clomipramine (*Swedo et al., 1989*; *Swedo, Lenane & Leonard, 1993*; *McGuire et al., 2014*). Recent studies also indicate that the glutamatergic neurotransmission modulator, N-acetyl-cysteine, is effective in alleviating the symptoms of TTM (*Grant, Odlaug & Kim, 2009*). Other drugs, such as atypical antipsychotic olanzapine, have shown success in placebo-controlled trials (*Van Ameringen et al., 2010*).

Neuroimaging studies indicated increased gray matter volume in striatum, amygdalo-hippocampal formation, and multiple cortical regions (including the cingulate, supplemental motor cortex, and frontal cortex) of patients with TTM (*Chamberlain et al., 2009*). This increase in gray matter was associated with a relatively decreased activity in cortical structures, and hyperactivity in the striatal region (*Fineberg et al., 2010*). Despite these findings, the neurobiology of TTM is not completely understood.

Animal models for the investigation of neurobiological aspects of TTM, as well as the search for new therapies, rely on the observation of barbering behaviors (BB). In laboratory and captive animals, BB is characterized by pulling or trimming of fur or whiskers (*Garner et al., 2004*). Some transgenic strains, *e.g.,* SAPAP3 and SLITRK5 knockouts, display compulsive/repetitive behavior, like excessive grooming (*Welch et al., 2007*; *Shmelkov et al., 2010*), however no specific BB has been described for them. Interestingly, SAPAP3 is a structural protein crucial for the proper docking of glutamatergic receptors (*Welch et al., 2007*).

In the present study, we describe the BB observed in NOS2 KO mice, and the effects of chronic treatment with clomipramine or memantine (a noncompetitive antagonist of NMDA receptors), as well as the partial rescue of NOS2 expression in haploinsufficient mice.

## METHODS

### Animals

Male and female NOS2 knockout (NOS2 KO; matrixes purchased from Jackson Lab # 2596, Bar Harbor, ME, USA), rederived into C57BL6/j background (WT, used as controls), were bred in the colony established in the Department of Genetics—Ribeirão Preto School of Medicine at the University of São Paulo. Unless otherwise stated, the adult animals (8 weeks old) were transported to the animal facility of the Department of Pharmacology one week prior to the beginning of experiments. The animals were housed in groups (3–6/cage)

**Table 1  Score for barbering behavior.** Based on *Garner et al. (2004)*.

| Area affected | Score |
|---|---|
| No hair loss | 0 |
| Loss of whiskers | +1 |
| Hair loss in snout | +1 |
| Hair loss in eye area | +1 |
| Hair loss in forehead | +1 |
| Hair loss in chest/neck | +1 |
| Hair loss in back | +1 |

and undisturbed (except for normal cage cleaning) with food and water available *ad libitum* except during experimental procedures. All efforts were made to minimize animal suffering. All protocols were approved and are in accordance with international guidelines for animal experimentation (146/2009).

## Drugs

Clomipramine hydrochloride (clomipramine; Sigma-Aldrich, São Paulo, Brazil) and memantine hydrochloride (memantine; Sigma-Aldrich, São Paulo, Brazil) were used. All drugs were dissolved in the drinking water (memantine at 0.006 mg/ml and clomipramine at 0.06 mg/ml concentration). The solutions were changed every two days, and the amount of liquid consumed was determined by changes in the bottle weight. The total volume was divided by the number of animals in each cage, and an average of 4.9 ml/day/animal was stably observed throughout the experiments.

## Actimeter test (ACTM)

The actimeter test was performed to evaluate general locomotor activity as well as repetitive behaviors. The detection of repetitive behavior relies on a specific pattern of interruption of infrared beams and comprises stereotypies and grooming. Briefly, the first beam is interrupted, followed by two interruptions of the adjacent one while the first beam remains obstructed. This apparatus consisted of two frames of 32 infrared beams in a square arena (Panlab-Harvard Apparatus, Barcelona, Spain). The animals were allowed to explore the arena for 15 min.

## Barbering behavior (BB)

The animals were scored regarding the state of the whiskers and fur, hair trimming or plucking. The score (Table 1) was based on Garner and colleagues (*2004*) body map for BB. The sum score for the absence of whiskers/fur in snout, eye surroundings, forehead, chest, neck, and back was determined every five days by an observer blind to the drug treatment.

## Cell culture, transfection, differentiation, and determination of MAP2 levels

PC12 cells, clone 615 from ATCC, were maintained in DMEM medium, supplemented with 10% fetal bovine serum (FBS) and 5% horse serum (HS), containing penicillin/streptomycin and glutamine. For the experiments, the cells were transferred to 12-well plates,

and 24 h later, the cultivation medium was changed to DMEM 0.5% FBS at a density of $5 \times 10^5$ cells/well. Twenty-four hours later, the cells were transfected with a mixture of lipofectamine 2.5% in OptiMEM medium and 2 ug of plasmids of 4 different interference sequences (shRNA) for NOS2 (0.5 μg of each sequence; #TR709254; Origene, Rockville, MD, USA). Twenty-four hours after transfection, the cultivation medium was changed, and the cells were incubated with KCl (25 mM) for five days. The levels of microtubule-associated protein 2 (MAP2) were determined by direct ELISA. Briefly, the samples (120 μg of total protein) were incubated in transparent 96-well U-shaped plates overnight at 4 °C. Following blockade with 1% BSA in PBST, antibody against MAP2 (#AB5622; Millipore, Darmstadt, Germany) was incubated overnight at 4 °C. The development of color following incubation with anti-rabbit IgG-HRP and BM Blue POD substrate (Roche Diagnostics, Espoo, Finland) was stopped by 1M HCl and the color intensity was read at 450 nm. The sample readouts, discounting the blank values, were expressed as percentage of the control-group (scrambled).

## EXPERIMENTAL DESIGN

### Time course of drug treatment effect in BB

Independent cohorts of NOS2 KO male and female mice (8–9 weeks of age at the beginning of the experiment) were scored for BB and received clomipramine (10 mg/kg) for 20 days in the drinking water. The average amount of solution consumed by the animals was monitored every two days to correct the drug intake. Given that no difference was observed between males and females regarding the responsiveness to clomipramine, we opted to use only female mice in the subsequent experiments. Thus, a separate cohort of female NOS2 KO mice received memantine (1 mg/kg) for 20 days in the drinking water. The BB was scored at the start of the treatment and monitored every five days until the end of the experiment.

### Rescue of BB by partial expression of NOS2

NOS2 haploinsufficient animals were obtained by crossing of NOS2 KO and WT in two cohorts: first KO sire and WT dam, second WT sire and KO dam. Age matched pups from NOS2 KO parents were generated as controls. The level of BB in the offspring (males and females) was observed from the 2nd to 9th week-of-age.

### Repetitive behavior in NOS2 KO and WT mice

Female NOS2 KO and WT were submitted to the actimeter for 15 min. The total number of interruptions of the infrared beams was considered as total activity (*Casarejos et al., 2013*). The number of repetitive movements (including stereotypies and grooming) were also counted by the apparatus and normalized by the total activity, therefore expressed as percentage of repetitive behavior.

### Effect of clomipramine on NOS2 KO repetitive behavior

This experiment was designed to address if the clomipramine treatment also modifies other phenotypic characteristics of NOS2 KO, i.e., increased repetitive movements; as

well as a possible effect in animals' total activity. To this aim, female NOS2 KO received clomipramine (10 mg/kg/day, in the drinking water) or vehicle during 14 days. On day 13, half of the vehicle-treated group received clomipramine in the drinking water. Therefore, the following groups were obtained: vehicle, acute clomipramine, and repeated clomipramine.

### Role for NOS2 in activity-induced expression of MAP2 in PC12 cells

PC12 cells, transfected with NOS2 shRNA or scramble (non-coding sequence), were differentiated in 25 mM KCl for five days. At day 5, the cells were lysed and the levels of MAP2 determined by ELISA.

### Statistical analysis

Experiments were analyzed by one- or two-way ANOVA (considering drug treatment or genotype, and time as factors) or Student's $t$ test. The Fisher's LSD *post hoc* test was used when appropriate. Unless otherwise stated, all data is expressed as Mean/SEM. Values of $p < 0.05$ were set as significant.

## RESULTS

### Time course of drug treatment effect in BB

As shown in Figs. 1B–1D, there is a significant effect of clomipramine treatment on BB in both males [$F(1, 18) = 27.03$; $p < 0.0001$] and females [$F(1, 8) = 29.04$, $p = 0.0007$]. The *post hoc* analysis (Fisher's LSD) indicates a significant difference between clomipramine and vehicle treated groups at days 10, 15, and 20 for both males ($p < 0.0001$ for all) and females ($p = 0.0011$; 0.0038 and 0.0038, respectively). The treatment with memantine, only performed in females, was also effective on BB [$F(1, 12) = 38.69$, $p < 0.0001$], with significant difference from controls observed at days 10, 15 and 20 ($p = 0.0024$ for all).

### Rescue of BB by partial expression of NOS2

Analysis by two-way ANOVA indicates a significant effect of the genotype [$F(1, 105) = 99.87$, $p < 0.0001$], the age [$F(6, 105) = 34.14$, $p < 0.0001$], and interaction between these factors [$F(6.105) = 5.23$, $p < 0.0001$] on the barbering behavior exhibited by NOS2 KO compared to NOS2 haploinsufficient animals from weaning age, Fig. 1D. Fisher's LSD test indicates significant differences starting from the 4th until 9th week-of-age ($p < 0.05$). No difference was observed at weaning age (3rd week) between pups from WT and NOS2 KO mothers.

### Repetitive behaviors in NOS2 KO and WT mice

Student's $t$ test indicates a significant effect of genotype, with NOS2.KO females exhibiting reduced total activity [$t(11) = 4.79$, $p = 0.0006$]. However, an increase in the percentage of repetitive movements was observed [$t(11) = 2.69$, $p = 0.021$], as found in Figs. 2A and 2B.

### Effect of clomipramine on NOS2 KO repetitive behavior

As observed in Fig. 2, there was no significant effect of clomipramine treatment on (c) total activity in NOS2 KO females [$F(2, 14) = 0.43$, $p = 0.66$]. However, repeated, but not

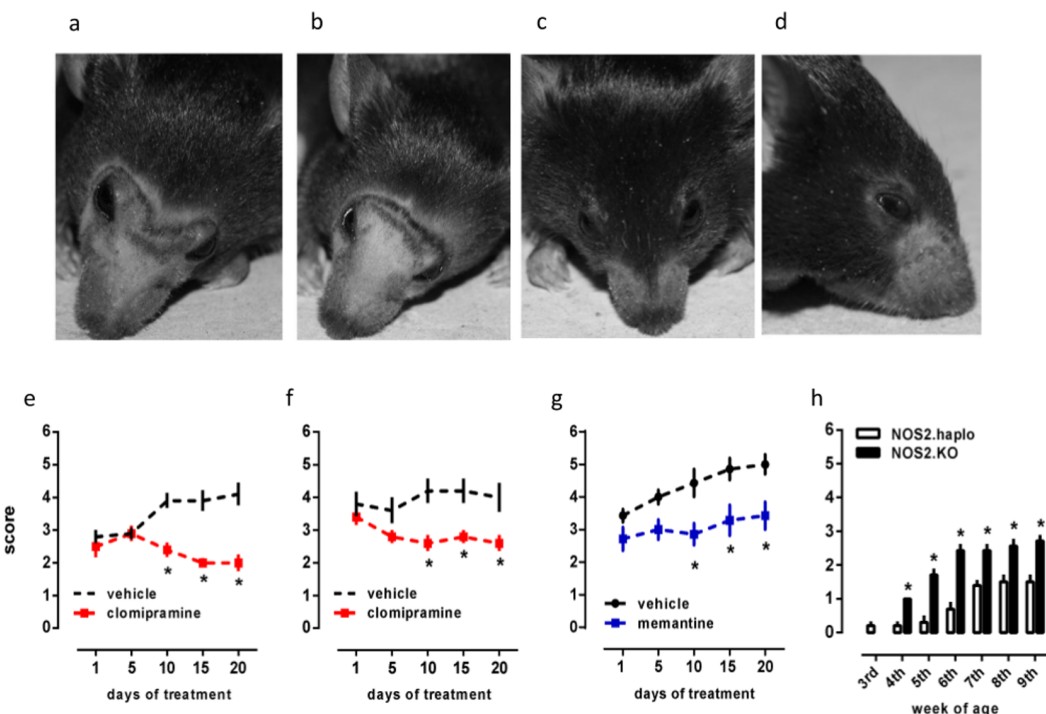

**Figure 1** **Rescue of barbering behavior in NOS2.KO.** (A) Phenotype of NOS2KO barbering behavior (BB), showing two vehicle- (A and B) and two clomipramine-treated (C and D) female NOS2KO. Effect of clomipramine (red squares; 10 mg/kg in the drinking water) on BB in males (E, $n = 10$/group) and females (F, $n = 5$/group) NOS2KO mice. (G) Effect of memantine (blue squares; 1 mg/kg in the drinking water) on BB in NOS2KO female mice ($n = 7$/group). (H) The partial rescue of NOS2 expression delays the onset and intensity of barbering behavior ($n = 7$–10/group). $*p < 0.05$ from vehicle or NOS2.haploinsufficient group at the same time point. Photo credit: Plinio Cabrera Casarotto.

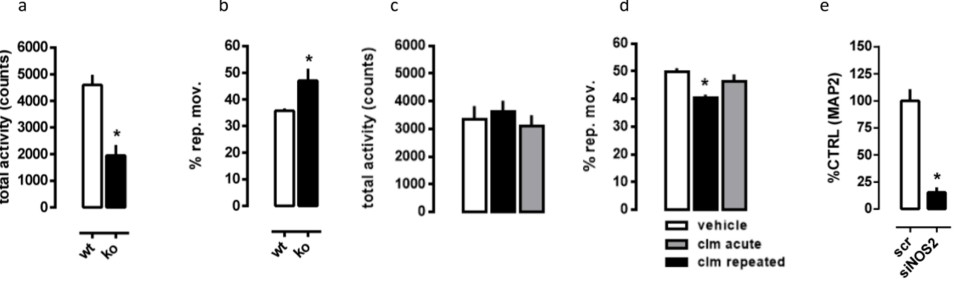

**Figure 2** **Repetitive and locomotor behavior of NOS2.KO and WT, and effect of siNOS2 in PC12 cells MAP2 expression.** Reduced total activity (A) and increased percentage of repetitive behaviors (B) in NOS2KO females compared to WT ($n = 6$–7/group). Effect of clomipramine (10 mg/kg in the drinking water) on total activity (C) and repetitive behaviors (D) in NOS2KO females ($n = 5$–6/group). (E) the silencing of NOS2 compromises the expression of MAP2 in KCl-induced differentiation of PC12 cells ($n = 4$–5/group). $*p < 0.05$ from WT, scrambled (scr) or vehicle-treated group.

acute, treatment with clomipramine decreased the percentage of repetitive movements (d) compared to the control group [$F(2, 14) = 6.18$, $p = 0.012$].

### Role for NOS2 on activity-induced expression of MAP2 in PC12 cells

The Student's $t$ test indicates a significant effect of NOS2 silencing on the levels of MAP2 in PC12 cells [$t(7) = 6.60$, $p = 0.0003$], as found in Fig. 2E.

## DISCUSSION

The present study suggests that the increased barbering behavior (BB) of NOS2 KO mice could be a putative model of trichotillomania (TTM). NOS2 KO exhibited increased BB after weaning (significantly higher at 4th week-of-age), and this phenotype increases until adulthood. TTM is an impulse control disorder, characterized by hair pulling or trimming. To our knowledge, there is no established animal model to study this disorder, which reflects in the poor pharmacotherapy and interventions available (*Johnson & El-Alfy, 2016*). In animals, BB is a rare phenotype, with an incidence of approximately 2.5% in laboratory mice (*Garner et al., 2004*). In our facilities, BB frequency was even lower (less than 0.5% out of 500 animals checked). However, all NOS2 KO animals exhibited some degree of BB. This phenotype was described in the manufacturer's webpage (https://www.jax.org/strain/002596), and all attempts to eradicate BB in the colony resulted in loss of the genotype.

One of the few pharmacological options to treat TTM is clomipramine, a tricyclic antidepressant drug that preferentially inhibits the serotonin reuptake system (*Swedo et al., 1989*; *Swedo, Lenane & Leonard, 1993*). In line with currently available data, our results indicate that repeated clomipramine administration was able to prevent the evolution of BB in NOS2 KO. Both males and females were equally responsive to the drug treatment. A possible confounding factor in the analysis of BB is the allobarbering instead of putatively selfbarbering. In the present case, we observed an incidence of 100% of BB in NOS2 KO in our colony; however, parts of the animal body only accessible by a conspecific, such as the back, were found barbered. This led us to conclude that the BB observed in these animals is a mixture of self- and allobarbering.

TTM is comorbid to several other psychiatric disorders, such as anxiety and obsessive-compulsive disorder (*Grant et al., 2017*). To investigate this association in NOS2 KO, the animals were submitted to the actimeter test, where both the total activity and the presence of spontaneous repetitive behavior were analyzed. Our results indicate a decreased activity of NOS2 KO associated with an increase in repetitive movements compared to WT animals. Interestingly, clomipramine was only effective in reducing the last parameter, with no effect on total activity. Despite reduced activity, this strain exhibits an antidepressant-like phenotype (*Montezuma et al., 2012*), and a deficit in extinction of conditioned fear (*Lisboa et al., 2015*), but no changes in repetitive behaviors was described. This mouse strain also presents decreased lung tumorigenesis induced by urethane (*Kisley et al., 2002*). On the other hand, the lack of NOS2 does not change the animal's survival rate to bacterial lipopolysaccharide (LPS) (*Laubach et al., 1995*).

Compulsive-like behaviors have been observed in many transgenic animals. For example, Francis Lee's group described increased grooming behavior in SLITRK KO mice (*Shmelkov et al., 2010*). Rodents lacking the adaptor protein SAPAP3 also presented increased grooming (*Welch et al., 2007*). Notwithstanding, this protein is responsible for anchoring glutamatergic receptors in the cell membrane and multiple rare missense variants in its gene sequence were found in OCD- and TTM-suffering patients (*Züchner et al., 2009*). Two other studies pioneered the analysis of BB in transgenic mice, and described increased grooming and BB in animals lacking the transcription factor HOXB8 (*Greer & Capecchi, 2002*; *Chen et al., 2010*). These two articles indicates that HOXB8 plays a major role in the excessive grooming phenotype, and is mainly expressed in components of CSTC. This is also an interesting possibility for NOS2. This enzyme level is highly modulated in cultured glial cells after several stimuli, such as LPS and interferon (*Galea, Feinstein & Reis, 1992*; *Garthwaite & Boulton, 1995*). Interestingly, the studies from Mario Capecchi's group points that the HOXB8 is expressed in bone-marrow-derived microglial cells. Thus in an intriguing speculative scenario, glial cells play a central role in the maturation and function of the CSTC components, leading to BB or excessive grooming. However, it is important to highlight that knocking out of NOS2 apparently does not lead to excessive grooming at the same extension of HOXB8 deletion. It is clear in the studies from *Greer & Capecchi (2002)* as well as in Chen and colleagues (*2010*) that the animals exhibited extensive lesions in the snout area. This feature was absent in the NOS2 KO, and the test used in the present study (actimeter) does not address if the increase in repetitive movements is a consequence of increased grooming behavior.

Our understanding of the neurobiology of TTM relies on few studies using brain image techniques suggesting a decrease in frontal cortical areas (mainly inferior frontal cortex) associated with an increase in striatal activity (*Chamberlain et al., 2009*; *Fineberg et al., 2010*). As previously mentioned, the cortico-striatal-thalamic circuitry (CSTC) is a series of reverberatory loops that integrate external signals and trigger motor programs in response. One of the integrative structures in this circuitry is striatum, including its multiple subdivisions, and disturbances in particular "hubs" in the CSTC are associated with different pathologies, from Huntington's disease to obsessive-compulsive disorder, including trichotillomania (for review see *Langen et al., 2011b*; *Langen et al., 2011a*). As a proof of concept, the treatment with the NMDA antagonist memantine was equally effective as clomipramine, in preventing the evolution of BB in NOS2 KO mice. Taken together, our results suggest a potential therapeutic approach for TTM with compounds acting on the glutamatergic system. In fact, drugs acting on decreasing glutamatergic neurotransmission have been tested in TTM patients successfully. For example, N-acetyl-cysteine was found to be effective in alleviating TTM symptoms in both double blind placebo-controlled and case studies (*Coric et al., 2007*; *Grant, Odlaug & Kim, 2009*; *Rodrigues-Barata et al., 2012*). This compound also exhibited promising results in OCD suffering patients (for review see *Oliver et al., 2015*). Riluzole, another compound with antiglutamatergic properties, was also found effective in OCD-related disorders (*Emamzadehfard et al., 2016*).

Data from literature linking NOS2 to putative neuronal developmental processes that could lead to malfunctions in CSTC is also scarce. One of the few pieces of evidence

about this topic was provided by Arnhold and colleagues (*2002*) describing the role of NOS2 in the early stages of neuronal maturation. They depicted a functional part for this enzyme in the cell differentiation and migration in embryonic cortex, although not through the canonical pathway of NO acting on soluble guanylate cyclase. Cultured cells from E14 rat embryo, when incubated with selective NOS2 inhibitors, showed a lower number of microtubule-associated protein (MAP2)-labeled neurons. Similarly, our data indicates that silencing NOS2 expression, leads to a reduction in the levels of MAP2 in activity-dependent differentiation of PC12 cells *Banno et al. 2008*. The role of NO in activity-dependent differentiation of PC12 cells has been described (for example *Nakagawa, Yoshida & Miyamoto, 2000*); however, the NOS1 isoform is implicated as being responsible for this process. This discrepancy may be due to the culture conditions between the studies. For example, Nakagawa and colleagues used a serum-free medium and 45 mM of KCl to differentiate the cells, while we rely on 0.5% FBS and 25 mM of KCl. Moreover, in the former study the authors cultivated a low density of PC12 cells (approximately 50 cells/cm$^2$; in a 96-well plate), while we used a high density from the beginning of the experiment—125,000 cells/cm$^2$ (in a 12-well plate).

## CONCLUSION

Based on our data and the evidence in literature, we speculate that increased BB observed in NOS2.KO is putatively a resultant of inefficient inhibition of striatal motor programs by PFC. The lack of NOS2 in the early stages of cortical development could affect the maturation of cortical neurons, which in turn would lead to the described phenotype. The BB observed in these animals is sensitive to treatment with clomipramine and memantine, strengthening its role as a putative model of TTM. Further experiments will be necessary to clarify the role of NOS2 in the development of the BB phenotype and its functional consequences in the CSTC; as well as validate its sensitivity to other drugs.

## ACKNOWLEDGEMENTS

The authors thank to Eleni T. Gomes (USP), Outi L. Nikkilä and Sulo J. Kolehmainen (UH) for their technical assistance.

### Funding

This study was supported by grants from Fundação de Amparo à Pesquisa do Estado de São Paulo—Fapesp (2011/02746-4 and 2012/17626-2), Conselho Nacional de Desenvolvimento Científico e Tecnológico—CNPq (471382/2011-6) and the European Research Council (iPlasticity, 322742). The funders had no role in study design, data collection and analysis, decision to publish, or preparation of the manuscript.

### Grant Disclosures

The following grant information was disclosed by the authors:

Fundação de Amparo à Pesquisa do Estado de São Paulo—Fapesp: 2011/02746-4 and 2012/17626-2.
Conselho Nacional de Desenvolvimento Científico e Tecnológico—CNPq: 471382/2011-6.
European Research Council: iPlasticity, 322742.

## Competing Interests

The authors declare there are no competing interests.

## Author Contributions

- Plinio C. Casarotto conceived and designed the experiments, performed the experiments, analyzed the data, prepared figures and/or tables, approved the final draft.
- Caroline Biojone and Francisco S. Guimaraes conceived and designed the experiments, analyzed the data, authored or reviewed drafts of the paper, approved the final draft.
- Karina Montezuma performed the experiments, analyzed the data, approved the final draft.
- Fernando Q. Cunha and Samia R.L. Joca contributed reagents/materials/analysis tools, approved the final draft.
- Eero Castren contributed reagents/materials/analysis tools, authored or reviewed drafts of the paper, approved the final draft.

## Animal Ethics

The following information was supplied relating to ethical approvals (i.e., approving body and any reference numbers):

*In vivo* experiments were conducted in conformity with the Ribeirão Preto School of Medicine, University of São Paulo local Ethical Committee (protocol 146/2009), in conformity with ARRIVE guidelines for animal experimentation.

## Data Availability

Figshare: https://figshare.com/projects/NOS2_knockout_mice_as_a_model_of_trichotillomania_/27259.

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
