# Peer review of "Inducible nitric oxide synthase (NOS2) knockout mice as a model of trichotillomania"

_PeerJ, doi:10.7717/peerj.4635_

## Round 0.1 · original submission · Minor Revisions

Please address as many suggestions as possible as they will greatly improve the impact of the manuscript

Reviewer 1 ·

Basic reporting

In this paper, the authors reported the phenotype of a inducible nitric oxide synthase (NOS2) knockout mice. They found that this mutant mouse exhibited barbering behavior (BB) which might be used as a model of trichotillomania (TTM). Furthermore, they showed that two drugs, clomipramine and memantine, could partially rescue the BB phenotype of NOS2 knockout mice. They also found a reduction of MAP2 in the NOS2 knockdown PC12 cells during activity-dependent differentiation. This study is interesting in regard to a novel model of TTM. However, lacking of proper controls and evidences make me having a hard time to evaluate this work.
1. Chen SK et al. Cell, 2010 and Greer JM et al. Neuron, 2001 showed that Hoxb8 mutant mice exhibited compulsive grooming and hair removal which were similar to TTM. They also found this behavior disorder was associated with mutant microglia. The authors should cite, discuss and learn from those two papers.
2. The authors mentioned in line 84-86 (Introduction) that some mice display excessive grooming without BB. Could the authors explain the differences between BB and excessive grooming? Are there any literatures that describe the differences?
3. Since the authors used the NOS2 knockout mice from Jackson Lab, they should introduce the phenotypes that were identified by previous studies. Actually, NOS2 mutant mice have a series of phenotypes, including abnormal inflammatory responses and have been widely used in several fields. How could the authors rule out the possibility that the BB phenotype was just a side effect of other defects? It is possible the BB phenotype in this study was not induced by the change in the nervous system.
4. Without any evidences to indicate the efficiency of NOS2 knockdown in PC12 cells, it is hard to evaluate the MAP2 data. Even if it's true, could the authors explain the relationship between the MAP2 data with the BB phenotype? I think it is inappropriate to use the indirect and preliminary evidence in this study to draw such a conclusion in line 288-290. The reduction of MAP2 and BB could be two parallel phenotypes without any interaction. Meanwhile, the authors should culture the primary neurons of NOS2 knockout mice and analyze the phenotype in vitro.
5. Figure 2: In 2A, the value of KO is about 2000. Meanwhile, in 2C, the vehicle (also KO) is over 3000. Could the author explain this huge difference. Moreover, the authors should include wt mice (with or without drug treatment) in the 2C and 2D.

Experimental design

no comments

Validity of the findings

no comments

Additional comments

no comments

·

Basic reporting

The authors describe the NO knockout mouse as a potential mouse model of trichotillomania. In general, the manuscript is well written, the rational for the study is described well, the methods are described adequately, the results are presented well, and the interpretation of results is well supported and presented in a context that is supported by publications in the field.

Experimental design

The experimental design is appropriate that answers a well defined question, which is relevant and meaningful. The number of animals per group was as low as 5, which is minimal as generally 12 animals per experimental group is a minimum in my research laboratory to have enough statistical power. Nevertheless, most of the results appeared to have robust statistical results.

Validity of the findings

The data is generally robust, although sample sizes are sometimes very small. The female and male data needs to be reanalyzed as described in the comments to the authors. Appropriate control groups were used. The interpretation of results are conclusions are appropriate for the results obtained. I especially like the authors cautious interpretation of their results without overstating their significance.

Additional comments

The authors describe the NO knockout mouse as a potential mouse model of trichotillomania. In general, the manuscript is well written, the rational for the study is described well, the methods are described adequately, the results are presented well, and the interpretation of results is well supported and presented in a context that is supported by publications in the field.

I would like to suggest just a few changes:

The barbering behavior is not well defined. The phenotype of the mice as shown in Fig. 1a could be due to selfbarbering or barbering by conspecifics in the home cage. Although the authors mention that some of the barbering on the back may be due to allobarbering, I would expect to see some evidence that the hair loss is due to selfbarbering. Whisker pulling and fur pulling on the snout, around the eyes and on the forehead can be due to allobarbering alone, as I have witnessed in my own mouse colony. Therefore, the authors cannot be sure that the barbering is due to selfbarbering unless they have observed the animals closely. Allobarbering could be due to expression of dominance by a conspecific. Do the authors have videos of animals pulling out their own whiskers or fur, or have they observed this? If the barbering is predominantly due to allobarbering, a link with compulsive-like behavior and trichotillomania would be more difficult to make as the phenotype could be due to expression of dominance.

For the statistical analyses with females and males, the authors should include an effect of sex and appropriate interaction effects in their ANOVAs, instead of testing females and males in separate ANOVAs or t tests. This will reveal whether the females and males responded similarly, or whether there were sex or interaction effects. Please include the true p value, instead of stating p<0.05 for every statistical analysis. This will provide more confidence in the overall results as some sample sizes are pretty small.

In the rescue of BB results section pups were used, but this was not described in the methods section. Please add a section for pups in the methods section and described at what ages they were tested.

For those results for which only females were used, for clarity I would like the authors to indicate that the results were obtained with females. For example, in the repetitive behaviors results section, the authors could write in the first sentence “NOS2.KO females” instead of just “NOS2.KO”.

Also, where there is a significant interaction effect, please describe the results that explain the interaction effect. This could result in novel insights that would otherwise get lost.

I would like to commend the authors on a thorough discussion that did not overstate the interpretation of their results.

---

## Round 0.2 · Minor Revisions

The manuscript needs a more formal revision of the text. Several pieces of information are missing or reported wrongly.

Please see below few examples:

"Drugs: clomipramine hydrochloride (clomipramine; Sigma-Aldrich, #C7291) and memantine nhydrochloride (memantine; Sigma-Aldrich, #M9292) were used."

The correct version would be:

"Drugs: clomipramine (clomipramine hydrochloride; Sigma-Aldrich, CITY, COUNTRY) and memantine (memantine hydrochloride; Sigma-Aldrich, CITY, COUNTRY) were used."

We do not need the catalogue number for chemicals.

Similarly, there are several spelling errors, for example: not n-acetylcysteine but N-acetylcysteine.

The References need a complete revision: Journal names: with capitals; titles of articles: without titles

Overall the manuscript does need some work and should be very carefully checked before submission.

---

## Round 0.3 · accepted · Accept

The authors have sufficiently revised the text. The editorial office will be in touch if there are other queries.

#